# Assessing Effectiveness of Colonic and Gynecological Risk Reducing Surgery in Lynch Syndrome Individuals

**DOI:** 10.3390/cancers12113419

**Published:** 2020-11-18

**Authors:** Nuria Dueñas, Matilde Navarro, Àlex Teulé, Ares Solanes, Mònica Salinas, Sílvia Iglesias, Elisabet Munté, Jordi Ponce, Jordi Guardiola, Esther Kreisler, Elvira Carballas, Marta Cuadrado, Xavier Matias-Guiu, Napoleón de la Ossa, Joan Lop, Conxi Lázaro, Gabriel Capellá, Marta Pineda, Joan Brunet

**Affiliations:** 1Hereditary Cancer Program, Catalan Institute of Oncology-IDIBELL, ONCOBELL, Hospitalet de Llobregat, 08908 Barcelona, Spain; nduenas@iconcologia.net (N.D.); mnavarrogarcia@iconcologia.net (M.N.); ateule@iconcologia.net (À.T.); msalinas@iconcologia.net (M.S.); siglesias@iconcologia.net (S.I.); emunter@iconcologia.net (E.M.); clazaro@iconcologia.net (C.L.); gcapella@iconcologia.net (G.C.); mpineda@iconcologia.net (M.P.); 2Centro de Investigación Biomédica en Red de Cáncer (CIBERONC), Instituto Salud Carlos III, 28029 Madrid, Spain; 3Hereditary Cancer Program, Catalan Institute of Oncology, Badalona, 089016 Barcelona, Spain; asolanes@iconcologia.net; 4Department of Gynecology, Bellvitge University Hospital, 08908 Hospitalet de Llobregat, 089016 Barcelona, Spain; jponce@bellvitgehospital.cat; 5Department of Gastroenterology, Bellvitge University Hospital, Hospitalet de Llobregat, 08908 Barcelona, Spain; jguardiola@bellvitgehospital.cat; 6Department of General Surgery, Bellvitge University Hospital, Hospitalet de Llobregat, 08908 Barcelona, Spain; ekreisler@bellvitgehospital.cat; 7Department of Gynecology, Trias i Pujol University Hospital, Badalona, 089016 Barcelona, Spain; ecarballas.germanstrias@gencat.cat; 8Department of General Surgery, Trias i Pujol University Hospital, Badalona, 089016 Barcelona, Spain; cuadradin@gmail.com; 9Department of Pathology, Bellvitge University Hospital, Hospitalet de Llobregat, 08908 Barcelona, Spain; fjmatiasguiu.lleida.ics@gencat.cat; 10Department of Pathology, Trias i Pujol University Hospital, Badalona, 089016 Barcelona, Spain; napoleondelaossa@gmail.com; 11Department of Pathology, Hospital General de Catalunya—Grupo Quironsalud, 08203 Barcelona, Spain; 12Department of Pathology, Hospital del Mar Institute for Medical Research, 08003 Barcelona, Spain; lopgros@gmail.com; 13Hereditary Cancer Program, Catalan Institute of Oncology-IDBIGI, 17007 Girona, Spain

**Keywords:** Lynch syndrome, endometrial neoplasms, colorectal neoplasms, ovarian neoplasms, prophylactic surgical procedures, risk reduction, gynecological neoplasms, risk reducing surgery

## Abstract

**Simple Summary:**

Colorectal and endometrial cancers are the most important life-threating risk in Lynch syndrome subjects, with incidences at 75 years as high as 40–60%. However, surveillance has shown to be ineffective. Risk reducing surgeries are an option in Lynch Syndrome (LS) individuals to decrease incidence of this type of cancers. In this manuscript, we have analyzed the rates of colorectal and gynecological cancer in 976 LS individuals after a mean follow-up of 10.2 years (patients under regular surveillance or after a risk reducing surgery). We can confirm in the largest study published up to the present in a single-institution that risk reducing surgeries are effective in decreasing incidence of colorectal and gynecological cancer in all LS carriers. Moreover, is the first report showing a decrease in all-cause mortality cumulative incidence in females with Lynch syndrome that undergo gynecological risk reducing surgery.

**Abstract:**

Background: Colorectal (CRC) and endometrial cancer (EC) are the most common types of cancer in Lynch syndrome (LS). Risk reducing surgeries (RRS) might impact cancer incidence and mortality. Our objectives were to evaluate cumulative incidences of CRC, gynecological cancer and all-cause mortality after RRS in LS individuals. Methods: Retrospective analysis of 976 LS carriers from a single-institution registry. Primary endpoints were cumulative incidence at 75 years of cancer (metachronous CRC in 425 individuals; EC and ovarian cancer (OC) in 531 individuals) and all-cause mortality cumulative incidence, comparing extended (ES) vs. segmental surgery (SS) in the CRC cohort and risk reducing gynecological surgery (RRGS) vs. surveillance in the gynecological cohort. Results: Cumulative incidence at 75 years of metachronous CRC was 12.5% vs. 44.7% (*p* = 0.04) and all-cause mortality cumulative incidence was 38.6% vs. 55.3% (*p* = 0.31), for ES and SS, respectively. Cumulative, incidence at 75 years was 11.2% vs. 46.3% for EC (*p* = 0.001) and 0% vs. 12.7% for OC (*p* N/A) and all-cause mortality cumulative incidence was 0% vs. 52.7% (*p* N/A), for RRGS vs. surveillance, respectively. Conclusions: RRS in LS reduces the incidence of metachronous CRC and gynecological neoplasms, also indicating a reduction in all-cause mortality cumulative incidence in females undergoing RRGS.

## 1. Introduction

Lynch Syndrome (LS) is characterized by an inherited defect in the mismatch repair (MMR) genes ((*MLH1*, *MSH2*, *MSH6*, *PMS2*) or *EPCAM* gene deletions, resulting in silencing *MSH2* gene in epithelial tissues)). It is the first cause of inherited colorectal cancer (CRC) and endometrial cancer (EC) [1]. CRC cumulative incidences at 75 years by genes are 48.3–57.1%, 46.6–51.4%, 18.2–20.3% and 10.4% for *MLH1*, *MSH2*, *MSH6* and *PMS2* mutation carriers, respectively. Moreover, LS carriers have a high risk of developing multiple CRC and at a younger age [2,3]. Gynecological cancer risk is also increased. Endometrial cancer (EC) incidence is comparable to CRC, being the cumulative incidences at 75 years by genes 37, 48.9, 41.1 and 12.8% for *MLH1*, *MSH2*, *MSH6* and *PMS2* mutation carriers, respectively. Ovarian cancer (OC) risk is also significantly increased with cumulative incidences at 75 years of 11, 17.4, 10.8 and 3% for *MLH1*, *MSH2*, *MSH6* and *PMS2* mutation carriers, respectively. The modestly increased risk of CRC and gynecological cancer of *PMS2* mutation carriers is not evident before 50 years of age [2,3]. It should be noted, however, that the prognosis of Lynch associated tumors is generally good [4].

Follow-up with regular colonoscopies every 1–3 years demonstrated a significant reduction in CRC-related mortality in the LS population [5,6]. Of note, interval cancers (tumors diagnosed between scheduled colonoscopies) continue to be diagnosed, and shorter colonoscopy intervals have not resulted in a reduction in incidence, a lowering of stage or an improvement in 10 year survival [7,8,9,10]. To reduce the incidence of metachronous CRC, extended colectomy is considered an option as it is a safe procedure without a significant impact on quality of life despite altering bowel function [11,12]. Retrospective series and two meta-analyses reported a four-fold reduction in the incidence of metachronous CRC with no advantage in overall survival [13,14,15,16,17,18,19,20,21,22,23,24,25]. Most international guidelines recommend discussing the pros and cons of extended surgery with CRC affected LS individuals before performing colorectal surgery for a first neoplasm [26,27,28,29,30].

Screening for EC and OC is not well established. Patients are usually diagnosed due to their symptoms, even if they had a prior normal screening with pelvic ultrasound [31,32]. Even though endometrial biopsy is a better diagnostic tool, it is a badly tolerated procedure in women under surveillance for a long period of time [33]. Schmeler et al., showed that risk reducing hysterectomy and oophorectomy in females with LS reduces the incidence of gynecological cancer by 100%, without major surgical complications [34]. In this context, international guidelines recommend discussing risk reducing gynecological surgery (RRGS) at age 35–40 after completion of childbearing in *MLH1*, *MSH2* and *MSH6* carriers [26,27,28,32]. Doubts exist regarding *PMS2* carriers because of their lower cancer risk [3]. To our knowledge, no study has evaluated the impact of this procedure on mortality in LS subjects.

Our objective was to evaluate the outcomes of risk reducing surgery in a single-institution LS cohort. We have evaluated cumulative incidences of metachronous CRC and all-cause mortality comparing segmental surgery (SS) vs. extended surgery (ES), as well as cumulative incidences of EC, OC and all-cause, EC-specific and OC-specific mortality comparing RRGS with standard gynecological follow-up.

## 2. Results

Of all 976 carriers, 678 were diagnosed with at least one malignancy: in 410 (42.0%) the first neoplasms were CRC (384 colonic (39.3%) and 26 rectal (2.7%)), and 125 (12.8%) were gynecological tumors (97 endometrial (9.9%), 28 ovarian (2.9%)) (Table 1). Mean age at diagnosis of first cancer was 47.6 years (range 18–86).

### 2.1. Colorectal Cancer Cohort

#### 2.1.1. Incidence of Colorectal Cancer

Of all 976 carriers, the CRC cohort included 425 individuals with at least one previous CRC (55.8% men, 44.2% women); all of which were adenocarcinomas. Mean age at first CRC diagnosis was 47.6 years (range 18–86). While 312 of the 425 carriers (73.4%) had a single cancer, 113 (26.6%) had more than one CRC (42 (9.9%) synchronous CRC and 71 (16.7%) metachronous CRC). The mean lapse between the first and the second CRC was 10.5 years (range 5–18). No differences regarding metachronous and synchronous CRC were evident between *MLH1* and *MSH2* carriers (*p* > 0.5). No conclusions could be drawn for *MSH6*, *PMS2* or *EPCA*M mutation carriers because of the small number of patients included (Table 2).

Data relating to first surgery were available in 290 of the 425 cases: 29 subjects (10.0%) underwent ES (79.3% subtotal colectomies) at a mean age of 46.0 years (range 25–79). and 261 (90.0%) underwent SS (64.0% right hemicolectomy and 11.9% left hemicolectomy) at a mean age of 46.9 years (range 18–83). No differences were observed in the proportion of carriers of *MLH1*, *MSH2* and *MSH6* gene mutations regarding gender, age at diagnosis, mean age at surgery and stage of first CRC between surgery groups (*p* > 0.5) (Table 1 and Table 3, Appendix A). Cumulative incidence at 75 years of metachronous CRC was 12.5% in the ES group vs. 44.7% in the SS group (*p* = 0.04), signifying an 84% reduction in the risk of developing CRC when ES were performed (Figure 1A). Significant differences in the rate of metachronous CRC were observed: 1 out of 29 (3.4%) in the ES group vs. 62 out of 261 (23.8%) in the SS group (*p* < 0.0001). (Table 3). Metachronous CRC in the SS group were mostly stage I (30.6%) and II (32.3%). One patient in the ES group developed a metachronous stage I rectal cancer (Appendix A). 

#### 2.1.2. Colorectal Cancer Mortality

Follow-up for patients was 10.9 years (range 0–28) in the ES group vs. 14.7 years (range 0–47) in the SS group. Death occurred in 3 of 29 (10.3%) patients treated with ES vs. 73 of 261 (27.9%) treated with SS. All-cause mortality cumulative incidence was 38.6% in the ES group vs. 55.3% in the SS group, (*p* = 0.31) (Figure 1B).

### 2.2. Gynecological Cancer Cohort

#### 2.2.1. Incidence of Gynecological Cancer

A total of 159 gynecological cancers were diagnosed in 150 of the 531 females with LS (28.2%): 114 EC (76.0%), 27 OC (18.0%) and 9 synchronous EC and OC (6.0%). Mean age at diagnosis was 49.9 years (range 28–80). Histology was predominantly endometrioid (Appendix A). A higher percentage of gynecological cancer was identified in *MSH2* (39.7%) and *MSH6* (37.6%) carriers compared with *MLH1* (20.1%) (*p* < 0.05) (Table 4).

Sixty-six women (12.4%) underwent RRGS at a mean age of 49.1 years (range 36–72) (57 HBSO, 1 hysterectomy plus salpingectomy and 8 hysterectomies). In the RRGS group, 6 women (9.1%) were diagnosed with an incidental EC identified in the pathological analysis of the surgical specimen. All 6 individuals were asymptomatic and had normal screening prior to surgery. All cancers were non-metastatic and three of them were pTis. However, in two cases extended surgical procedure were required after the diagnosis of EC. On the other hand, 117 EC were diagnosed in 465 (25.2%) women in the non-RRGS group at a mean age of 50.1 years (range 28–80), mostly stage I (*n* = 57, 48.7%). However, 4 women had stage IV EC (3.4%). No OC were diagnosed in the RRGS group (0%) vs. 36 in the non-RRGS group (7.7%): 16 stage I, 2 stage II and 7 stage III. No stage IV OC were diagnosed (Table 5 and Appendix A). Cumulative incidence at 75 years of gynecological cancer was 11.2% in the RRGS group vs. 53.5% in the non-RRGS group (*p* < 0.001). Cumulative incidences at 75 years of each cancer were: 11.2% vs. 46.3% of EC (*p* = 0.001) and 0% vs. 12.7% (*p* N/A) of OC, in the RRGS and the non-RRGS groups respectively. This signifies a 74 and 100% risk reduction of developing EC and OC when RRGS were performed (Figure 2A,B).

#### 2.2.2. Gynecological Cancer Mortality

Follow-up for patients with and without RRGS was 8.7 years (range 0–43) and 10.4 years (range 0–45), respectively. Death occurred in 0 out of 66 (0%) women in the RRGS group vs. 98 out of 465 (21.1%) in the non-RRGS group, 11 due to EC (2.4%) and 6 due to OC (1.3%). All-cause mortality cumulative incidence was 0% in the RRGS group vs. 52.7% in the non-RRGS group (*p* N/A) (Figure 3). EC-specific mortality cumulative incidence was 0% in the RRGS group vs. 5.9% in the non-RRGS group (*p* N/A) and OC-specific mortality cumulative incidence was 0% in the RRGS group vs. 2.6% in the non-RRGS group (*p* N/A).

## 3. Discussion

Most international guidelines recommend discussing the pros and cons of risk reducing surgery in affected LS individuals with CRC, and also considering risk reducing gynecological surgery (RRGS) in asymptomatic females with LS at age 35–40 after completion of childbearing. However, the strength of the evidence supporting these recommendations is still weak due to the limited sample size. This work evaluating risk reducing surgery (extended surgery after a first CRC and risk reducing gynecological surgery) is the largest single-center study published up to the present and points that in LS population these strategies have an impact on decreasing the cumulative incidence of cancer at 75 years and the cumulative incidence of mortality.

As reported in the literature, LS subjects have a higher risk of developing one or more CRC with cumulative incidences by the age 75 years of 48.3–57.1% for *MLH1*, 46.6–51.4% for *MSH2*, 18.2–20.3% for *MSH6* and 12.8% for *PMS2* [2,3]. The reason for this increased incidence of CRC was initially explained by accelerated carcinogenesis [35], since performing colonoscopies every 1–3 years vs. no follow-up showed a reduction both in incidence and mortality in LS individuals [5,6]. However, recent studies are raising doubts regarding the secondary prevention of CRC with annual colonoscopies since no differences were found in the incidence of advanced CRC with respect to longer intervals of colonoscopies [7,8,9,10]. Moreover, some CRC have been proven to evolve from a non-polypous precursor lesion at MMR-deficient crypts via somatic mutations in the *CTNNB1* gene [36,37]. A recent publication identified differences in carcinogenesis between MMR genes, showing different risks of advanced adenoma (7.7% in *MLH1* vs. 17.8% in *MSH2* carriers) but a similar proportion of cancer (11.3% in *MLH1* vs. 11.4% in *MSH2* carriers) and different patterns of *CTNNB1* somatic mutations (50% in CRC from *MLH1* carriers vs. 7% in *MSH2* carriers), suggesting that CRC in *MLH1* mutation carriers may evolve more frequently through non-polypous precursor lesions and, therefore, are not prevented by colonoscopies [38].

Based on this increased risk of presenting both a first CRC and a metachronous CRC [2,3,4], it is important to discuss the extent of surgical resection for the first CRC. Several studies and two meta-analyses have reviewed this issue (Table 6) [13,14,15,16,17,18,19,20,21,22,23,24,25]. In the meta-analyses by Heneghan [16] and Anele, [17] the incidence of metachronous CRC was significantly lower in the ES group (6.8% and 6%) than in the SS group (23.5 and 22.8%) but no differences in 10 year survival were found. Similar to the published literature, cumulative incidence at 75 years of metachronous CRC in our series was lower in the ES than in the SS group (12.5 vs. 44.7%, *p* = 0.04). Moreover, all-cause mortality cumulative incidence was lower in the ES group (38.6%) than in the SS group (55.3%). However, this difference was not significant (*p* = 0.31). In agreement with previous reports [2,3], CRC in our series were diagnosed at younger ages in individuals carrying mutations in highly penetrant MMR genes (mean age: 45.2 and 46.7 years for *MLH1* and *MSH2* vs. 56.3 and 58.8 years for *MSH6* and *PMS2*, respectively). The rates of metachronous CRC were also higher in these individuals (15.5 and 23.2% in *MLH1* and *MSH2* vs. 10.0 and 5.9% in *MSH6* and *PMS2*, respectively) (Table 2). 

The 10 year follow-up results of the double-blind randomized CAPP2 trial have been recently published and the study shows that treatment with 600 mg aspirin daily vs. placebo decreases the incidence of CRC. The rates of CRC were 9% in the aspirin group vs. 13% in the placebo group (HR = 0.65; 95% CI 0.43–0.97; *p* = 0.035) with no significant differences in adverse events or compliance between intervention groups [39]. To date, no study has compared extended surgery vs. segmental surgery plus chemoprevention with aspirin, so currently we cannot conclude that one approach is better than the other.

Based on the latest evidence, extensive colonic surgery could be strongly recommended to *MLH1* and *MSH2* mutation carriers, but there are doubts regarding extending this recommendation to *MSH6* mutation carriers [30]. In our series, 30.3% of *MSH6* mutation carriers developed at least one CRC, and metachronous cancers were diagnosed in 0% of the individuals in the ES group vs. 16.7% in the SS group. For this reason, we recommend caution in these individuals, especially in those that develop CRC at younger ages. As for elderly subjects (>75–80) and rectal cancer, considering associated morbimortality, segmental surgery plus endoscopic surveillance seems the best option. Chemoprevention with aspirin has to be individualized in this situation, taking into account interactions with other comorbidities and treatments. In light of the literature published so far, this decision must be considered by a multidisciplinary team, discussing the pros and cons of both types of surgeries in every LS individual with a recent diagnosis of CRC. It is also important to highlight that ES does not completely prevent the risk of metachronous CRC, since rectal or sigmoid tissue is maintained to preserve functional outcome (as seen in one patient in our series). The risk of metachronous rectal cancer in these subjects has been reported as 3–12% at 10–12 years [4,14,18,40]. For this reason, regular endoscopic surveillance of the remaining colon or rectum should be maintained to reduce the incidence of CRC and its related mortality.

In females with LS, gynecological cancer risk has been proven to equal or exceed the risk of CRC and it can be the sentinel malignancy in up to 35–40% of cases, with cumulative incidences at 75 years of age of EC and OC of 37% and 11% for *MLH1*, 48.9 and 17.4% for *MSH2*, 41.1 and 10.8% for *MSH6* and 12.8 and 3% for *PMS2*, respectively [2,3]. Regarding EC, the results from our series also match those previously reported: gynecological tumors were diagnosed in 28.2% of the women and were the first neoplasms in 21.7% of women who developed cancer (Table 4 and Appendix A). However, the number of gynecological cancers in females carrying an *MSH6* mutation were lower than expected, probably because most of our patients were identified by clinical criteria (Amsterdam and Bethesda), which are focused mainly on CRC, which is less frequent than EC in *MSH6* carriers [2].

Our study confirms that in females with LS, RRGS is a beneficial procedure for reducing the incidences of EC and OC compared with regular follow-up. In our analysis of 531 women with LS, EC were significantly reduced when performing RRGS (25.2 vs. 9.1%) and no OC were diagnosed in the RRGS group (7.7 vs. 0%). It is worth mentioning that our series also includes 85 female carriers of a mutation in the *MSH6* gene, 32 of whom (37.6%) developed gynecological cancer (24 EC, 5 OC and 3 synchronous EC and OC), all from the non-RRGS group (Table 4 and Table 5). We replicated the observations of Schmeler et al. [34], that included 315 females with LS (138 *MLH1*, 175 *MSH2* and 3 *MSH6*); 61 of whom underwent RRGS at a mean age of 41 years. Incidences of EC and OC were similar to our study (Table 7). Even though the incidence of EC is considered 0% in the study by Schmeler et al., three females were incidentally diagnosed of EC at the time of prophylactic surgery and were included in the non-RRGS group. In our series, all the six cancers diagnosed in the RRGS group were incidental findings during the pathological analysis of the surgical specimen and two of them required extended oncological surgery. We have maintained these six diagnoses in the RRGS group because we consider that it better reflects routine clinical practice and is a more accurate consideration regarding the risks and benefits of RRGS. The mean age at RRGS in our series was 49.1 years, older than expected, since in our institution RRGS were clearly recommended after 2015 and was then offered to women under surveillance. Older age at RRGS could explain the higher incidence of EC in females in the RRGS group. No primary peritoneal cancer was identified in our analysis. The magnitude of the risk of primary peritoneal cancer in females with LS is unknown, but it is probably low as only five cases have been described so far [41].

Our present estimates, show a reduction in all-cause, EC-specific and OC-specific mortality cumulative incidence in the RRGS group (all-cause mortality of 0 vs. 52.7%, EC-specific mortality of 0 vs. 5.9% and OC-specific mortality of 0 vs. 2.6% for RRGS and non-RRGS group, respectively). It must be taken into account that oncologic treatments have evolved since the start of this study, so it is possible that survival in females with LS diagnosed with gynecological cancers will improve in the future.

So far, screening options for EC include TVUS, endometrial sampling and Ca125, even though most women are diagnosed due to their clinical symptoms. Therefore, EC surveillance does not imply neither a reduction in stage at diagnosis nor a survival improvement. Moreover, surveillance for OC has proven to be unsuccessful [32,33]. In consequence, main international clinical guidelines recommend considering prophylactic hysterectomy with or without bilateral salpingoophorectomy around the age of 35–40 years and after fulfilling childbearing in *MLH1* and *MSH2* carriers, and later (at 40 years) for *MSH6*. To avoid early menopause complications, different approaches can be considered: oestrogenic hormonal replacement therapy (HRT) until the natural age of menopause, or a two-step surgery with hysterectomy after childbearing is complete followed by salpingoophorectomy after menopause [32]. More data are required to extend this recommendation to *PMS2* carriers because of their lower gynecological cancer risk, which is not increased before 50 years of age [2,3]. Nevertheless, it seems reasonable that if RRGS is considered, it can be offered after the natural age of menopause [26,27,28,32]. In line with these results, a recently published survey conducted by the prospective Lynch Syndrome database (PLSD) show that most of the referral centers included worldwide (91–95%) offer RRGS to carriers of pathogenic variants in *MLH1*, *MSH2* and *MSH6* but less (67%) to carriers of pathogenic variants in *PMS2*. Most of the centers (71%) also recommend oestrogen-only HRT between 35-55 years, approximately [42].

Important strengths of our analysis are the internal robustness due to the use of prospectively acquired data and the long-term follow-up and that all patients were treated in one single comprehensive cancer institution with the same protocol. Furthermore, this is the first report analyzing the impact of both colonic and gynecological risk reducing surgeries in the same LS series. Here, we describe for the first time a decrease in all-cause, EC-specific and OC-specific mortality cumulative incidence after RRGS. We were not able to calculate statistical significance in the cumulative incidence of mortality in the gynecological cancer cohort, because no woman died in the RRGS group after 8.7 years of mean follow-up vs. 98 women in the non-RRGS group after 10.4 years of mean follow-up.

We have calculated the cumulative incidence of mortality (all-cause, EC-specific and OC-specific) using chronological age and not time-on-study as the time scale, because we considered that the follow-up 5 or 10 years after a preventive surgery performed at a mean age of 45–50 years is not usually a long enough follow-up time and, furthermore, it does not take into account the age of the patient to calculate mortality [43,44]. Even more so, if we consider that, in general, the prognosis for these tumors is considered to be good [2,3]. However, this approach can be argued as unadjusted age scale can lead to biases.

In the CRC cohort, information about cause of death was lacking or it was not reliable in a high number of patients and only all-cause mortality cumulative incidence could be analyzed. It would have been interesting to analyze the cumulative incidence of metachronous CRC-specific mortality. However, considering the study follow-up time and the improvements in OS and DFS of the new combinations of oncologic treatments over the last years, it is possible that if differences in mortality were found, they were due to new treatment of the metachronous CRC and not surgical treatment of the first CRC.

A possible limitation of our study is that patients were allocated by the surgeon to segmental or extended surgery groups based on clinical criteria (age at CRC diagnosis, localization of the tumor, existence of synchronous cancer, patient’s preference, etc.) without information regarding LS condition, MMR protein expression or BRAF mutation in tumor. We can state, however, that there were no statistically significant differences between groups regarding gender, mutated gene, age at CRC diagnosis or CRC stage (Table 1, Table 2 and Table 3 and Appendix A). A similar situation occurred in the gynecological cancer cohort, where individuals were included in each group based on patient’s preference, family history and existence of a benign pathology that required gynecological surgery and, after 2015 it was then offered to all females with LS. We verified that no significant differences existed between mean age at gynecological surgery or proportion of carriers for each gene (Table 1, Table 4 and Table 5). It must be considered that biases can also exist regarding differences between time of follow-up and number of patients included in each surgery group, both in the colonic and gynecological cohort.

Another possible limitation is the existence of a cohort effect, as patients were included in the study between 1969 and 2016 and during these years, life expectancy has increased and oncological treatments, surgical procedures and endoscopic techniques have improved, achieving improvements in diagnoses and increasing patients’ survival. Therefore, we must bear in mind that, due to the small number of patients and the nature of retrospective studies, certain selection biases and risk overestimations cannot be ruled out. For that reason, although risk reducing surgery has shown a tendency to reduce mortality in LS individuals, these results are not statistically significant and must be confirmed in prospective studies.

Another limitation of our study is that neither quality of life nor morbidity was evaluated. Other similar publications did not find important differences in generic quality of life in ES, unless total colectomy with ileorectal anastomosis was performed [11,12]. Two studies evaluating risk-reducing HBSO in females with LS concluded that surgery does not have a negative effect on quality of life, especially in those using hormonal therapy [45,46]. Any significant differences in surgical morbidity of CRC and gynecological surgery were not reported in several prospective studies [11,12,45,46]. It should be considered that if a 35–40-year female with LS who has fulfilled childbearing has to be intervened for a CRC, HBSO should be offered in the same surgical act to avoid further complications [32].

## 4. Materials and Methods 

### 4.1. Study Sample

This is a retrospective cohort study from the Catalan Institute of Oncology Hereditary Cancer Program that included 976 individuals (531 women, 445 men) belonging to 234 LS families: 826 proven carriers (84.6%) and 150 obligate carriers (15.4%) of a pathogenic mutation in one of the MMR genes (480 *MLH1*, 262 *MSH2*, 165 *MSH6*, 48 *PMS2* and 21 *EPCAM*). All the families were visited at the Genetic Counselling Units of our center between 1999 and 2016. Mean follow-up was 10.2 years (range 0–47) (Table 1). All patients underwent appropriate genetic counselling prior to all genetic tests and gave informed consent for genetic analysis and internal Ethics Committee approved this study. This study was performed in accordance with the Declaration of Helsinki.

### 4.2. Data Collection

Data supporting the results are stored in the clinical database of the Hereditary Cancer Program. If surgery was performed, information regarding indication of surgery (prophylactic vs. non-prophylactic), extension of surgery, date of surgery and pathological note was recorded. Information about surgery and pathological findings were verified by medical reports. Demographic, personal characteristics, genogram and personal history of cancer were collected. When missing data occurred, the subject was excluded from the analysis that involved the missing data point, but the subject was included in other analysis where complete data were available. Clinical information before LS diagnosis was reviewed retrospectively and included in the clinical database, while analyzed observations after the identification of a MMR gene mutation were prospective. After LS diagnosis, colonic and gynecological follow-up or RRGS was indicated. Follow-up recommendation in our institution included a 2-year interval colonoscopy with or without chromoendoscopy, beginning at 20–25 years (*MLH1* and *MSH2* carriers) or at 25–30 years (*MSH6* and *PMS2* carriers) unless an earlier onset cancer existed in the family. After 40 years of age, colonoscopy was performed annually. In women, yearly gynecological examination with transvaginal ultrasound (TVUS) was recommended after 30 years of age.

### 4.3. Colorectal Cancer Cohort

In the CRC cohort, all patients had at least one previous CRC. Colorectal surgeries for first cancers were decided by surgeons at diagnosis, considering clinical criteria (age of the subject, localization of the tumor and existence of a synchronous cancer). At that moment, most of the patients were not diagnosed with LS as LS screening in the colectomy specimen based on Jerusalem criteria was implemented in our center at the beginning of January 2016. Data about first CRC surgery were available in 290 of the 425 patients. Only the surgical treatment for the first CRC was considered. Information on the cause of death was missing in a high number of patients or it was unreliable, so this information was not considered. Each individual was counted only once, regardless of how many cancers he or she might have had. Cumulative incidence at 75 years of metachronous CRC and all-cause mortality cumulative incidence were compared between SS and ES. For cumulative incidence at 75 years of metachronous CRC endpoint, individuals were followed until diagnosis of metachronous CRC or censoring at death or date of last assessment. For all-cause mortality cumulative incidence endpoint, individuals were followed until death or date of last assessment.

### 4.4. Gynecologicall Cancer Cohort

In the gynecological cancer cohort, all women diagnosed with LS underwent gynecological screening with a yearly gynecological consultation that included anamnesis, physical examination and transvaginal ultrasound, starting at 30 years of age. Since 2003 RRGS were recommended to females with LS when there existed a family history of gynecological cancer or they presented symptoms caused by benign lesions. However, this surgery has been actively recommended since 2015, following international guidelines [27,28,32]. Data about gynecological cancer, gynecological surgery and cause of death were available in all the individuals. Cumulative incidence at 75 years of gynecological cancer (endometrial or ovarian) and all-cause, EC-specific and OC-specific mortality cumulative incidences were compared among females who had undergone RRGS and those who had not. For cumulative incidences at 75 years of EC and OC endpoint, individuals were followed until diagnosis of EC, OC, RRGS or censoring at death or date of last assessment. For all-cause, EC-specific and OC-specific mortality cumulative incidence endpoint, individuals were followed until death or date of last assessment.

### 4.5. Definitions

Segmental surgery for CRC was defined as right or left hemicolectomy, sigmoid colectomy, transverse colectomy, abdominoperineal resection or low anterior resection. Extended surgery was defined as total colectomy with ileorectal anastomosis or subtotal colectomy with ileosigmoid anastomosis. Synchronous CRC was defined as a CRC identified at the same moment or during the 6 months following the diagnosis of a primary CRC. Metachronous CRC was defined as a CRC diagnosed at least 6 months after first CRC diagnosis.

Females with LS were classified depending on the treatment received: RRGS vs. non-RRGS. RRGS was defined as surgery in a women not previously diagnosed with gynecological cancer. Surgery was either hysterectomy, hysterectomy plus salpingectomy or hysterectomy plus bilateral salpingoophorectomy (HBSO), whichever was considered best in a gynecological assessment. All women had a presurgical evaluation consisting of a transvaginal ultrasound. In case of suspicion, endometrial biopsies and/or complementary imaging examinations were performed. All-cause mortality considered any cause of death. EC-specific mortality and OC-specific mortality calculations considered women with stage IV EC or OC and whose death were attributable to progression of these neoplasms.

### 4.6. Mutation Testing

Patients referred for MMR mutation analysis were suspected of having LS because they fulfilled LS clinical criteria (Amsterdam or revised Bethesda guidelines) or, as of January 2016, Jerusalem criteria [47]. Point mutation analyses of MMR genes were performed by PCR amplification of exonic regions and exon-intron boundaries followed by Sanger sequencing or by next generation sequencing. Genomic rearrangements in MMR genes were analyzed by multiplex ligation dependent probe amplification kits (MRC-Holland). Variant classification was completed according to Insight guidelines [48]. Laboratory reports identifying a pathogenic variant in any MMR gene were required for confirmation. Untested individuals were considered obligate MMR mutation carriers whenever the evidence indicated that they were the only possible transmitters of the mutation to their offspring.

### 4.7. Statistical Analysis

The Kaplan–Meier method was used to estimate cumulative incidence at 75 years of cancer and all-cause mortality cumulative incidence, EC-specific mortality cumulative incidence and OC-specific mortality cumulative incidence. Age rather than time-on-study was used as the time scale, as recommended [43,44]. Comparisons of proportions between groups were assessed using the test for equality of proportions. Log-rank and Peto–Peto test compared survival curves between the different surgical techniques. To control for confounding variables, adjustments for specific mutated MMR gene were performed. In all tests, the level of statistical significance was set at *p* < 0.05. Statistical analyses were performed using R version 3.2.1 and 3.4.0.

## 5. Conclusions

In conclusion, this study confirms that colonic and gynecological risk reducing surgeries are effective decreasing the incidence of metachronous CRC and gynecological cancer in LS patients. This benefit was seen in all LS subjects; however, caution is still needed for *MSH6* and *PMS2* mutation carriers. This is the first report pointing to a reduction in the all-cause mortality cumulative incidence in females with LS that undergo RRGS.

## Figures and Tables

**Figure 1 cancers-12-03419-f001:**
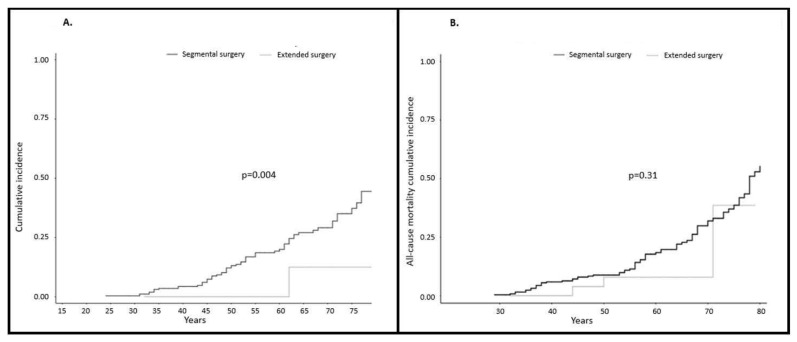
Colorectal cancer cohort: (**A**) Cumulative incidence at 75 years of metachronous colorectal cancer in Lynch syndrome subjects comparing extended surgery and segmental surgery: Cumulative incidence at 75 years of metachronous colorectal cancer was 12.5% for extended surgery vs. 37.3% for segmental surgery (*p* = 0.004); (**B**) All-cause mortality cumulative incidence in Lynch syndrome subjects comparing extended surgery and segmental surgery: All-cause mortality cumulative incidence was 38.6% for extended surgery vs. 55.3% for segmental surgery (*p* = 0.31).

**Figure 2 cancers-12-03419-f002:**
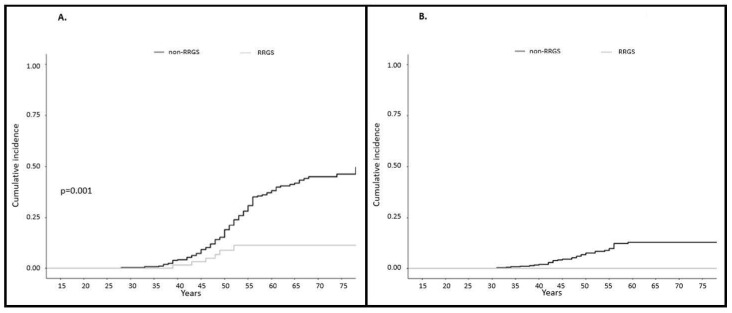
Cumulative incidence at 75 years of gynecological cancer in females with Lynch syndrome comparing risk reducing gynecological surgery and non-risk reducing gynecological surgery: (**A**) Cumulative incidence at 75 years of endometrial cancer. Cumulative incidence at 75 years of endometrial cancer was 11.2% for risk reducing gynecological surgery vs. 46.3% for non-risk reducing gynecological surgery (*p* = 0.001); (**B**) Cumulative incidence at 75 years of ovarian cancer. Cumulative incidence at 75 years of ovarian cancer was 0.0% for risk reducing gynecological surgery vs. 12.7% for non-risk reducing gynecological surgery (*p* = not assessable).

**Figure 3 cancers-12-03419-f003:**
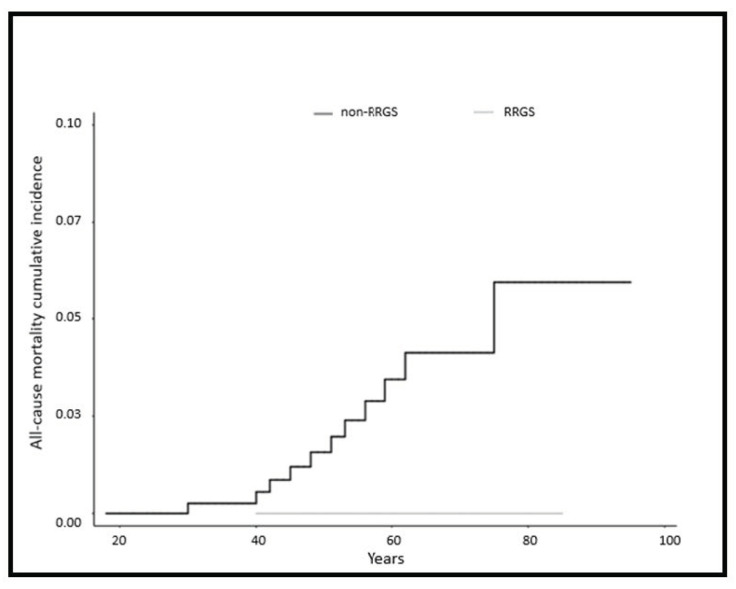
All-cause mortality cumulative incidence in females with Lynch syndrome comparing risk reducing gynecological surgery and non-risk reducing gynecological surgery: All-cause mortality cumulative incidence was 0.0% for 52.7% for risk reducing gynecological surgery vs. 52.7% for non-risk reducing gynecological surgery (*p* = not assessable).

**Table 1 cancers-12-03419-t001:** Patient characteristics.

	All Individuals	Colorectal Cancer Cohort		Gynecological Cancer Cohort	
		Extended Surgery	Segmental Surgery	RRGS ^1^	Non-RRGS ^2^
**TOTAL**	*n* = 976	*n* = 29	*n* = 261	*n* = 66	*n* = 465
**Sex**					
**Male**	445 (45.6%)	18 (62.1%)	146 (55.9%)		
**Female**	531 (54.4%)	11 (37.9%)	115 (44.1%)	66 (100%)	465 (100%)
**Mean age**	54.1 y ^7^ (18–95)	56.4 y ^7^ (32–79)	59.9 y ^7^ (29–96)	57.3 y ^7^ (40–85)	54.8 y ^7^ (18–95)
**MMR gene**					
***MLH1***	480 (49.2%)	19 (65.5%)	137 (52.5%)	33 (50.0%)	226 (48.6%)
***MSH2***	262 (26.8%)	5 (17.2%)	77 (29.5%)	19 (28.8%)	127 (27.3%)
***MSH6***	165 (16.9%)	1 (3.4%)	30 (11.5%)	10 (15.2%)	75 (16.1%)
***PMS2***	48 (4.9%)	1 (3.4%)	14 (5.4%)	3 (4.5%)	22 (4.7%)
***EPCAM***	21 (2.2%)	3 (10.3%)	3 (1.1%)	1 (1.5%)	15 (3.2%)
**Death**	221 (22.6%)	4 (13.8%)	41 (15.7%)	0 (0%)	98 (21.1%)
**Mean age at death (range)**	58.4 y ^7^ (25–89)	55.0 y ^7^ (44–71)	50.8 y ^7^ (29–84)		60.5 y ^7^ (25–89)
**First cancer diagnosis ^3^**	678 (69.5%)	29 (100%)	261 (100%)	33 (50.0%)	277 (59.6%)
**Colon**	384 (39.3%)	26 (89.7%)	214 (82.0%)	22 (33.3%)	119 (25.6%)
**Endometrial**	97 (9.9%)	1 (3.4%)	17 (6.5%)	4 (6.1%)	86 (18.5%)
**Ovarian**	28 (2.9%)	1 (3.4%)	3 (1.1%)	0 (0%)	25 (5.4%)
**Rectum**	26 (2.7%)	0 (0%)	19 (7.3%)	3 (4.5%)	9 (1.9%)
**Other GI ^4^**	26 (2.7%)	0 (0%)	2 (0.8%)	0 (0%)	10 (2.2%)
**Urologic ^5^**	14 (1.4%)	0 (0%)	0 (0%)	2 (3.0%)	5 (1.1%)
**Other non-LS**	103 (10.6%)	1 (3.4%)	6 (2.3%))	2 (3.0%)	23 (4.9%)
**Mean age at first cancer diagnosis (range)**	47.6 y ^7^ (18–86)	46.3 y ^7^ (25–79)	45.9 y ^7^ (18–83)	46.2 y ^7^ (28–66)	49.0 y ^7^ (18–86)
**Mean age at surgery of study (range) ^6^**		46.0 y ^7^ (25–79)	46.9 y ^7^ (18–83)	49.1 y ^7^ (36–72)	50.1 y ^7^ (28–80)

^1^*RRGS* Risk reducing gynecological surgery, ^2^
*non-RRGS* Non-risk reducing gynecological surgery, ^3^
*First cancer diagnosis* First neoplasm developed by the subjects of our series, sorted from more to less frequent, ^4^
*Other GI tumors* stomach, small bowel and bile ducts, ^5^
*Urologic tumors* bladder and urinary tract, *y* years, ^6^
*Mean age at surgery of study* Mean age at first colorectal cancer surgery (Extended surgery or segmental surgery) for the CRC cohort and mean age at gynecological surgery (risk reducing gynecological surgery or surgery for gynecological cancer) for the gynecological cancer cohort, ^7^
*y* years.

**Table 2 cancers-12-03419-t002:** Number of colorectal cancer and type of second colorectal cancer according to mutated genes.

Characteristics	TOTAL *n* = 425 (100%)	*MLH1**n* = 239 (56.2%)	*MSH2**n* = 112 (26.4%)	*MSH6**n* = 50 (11.8%)	*PMS2**n* = 17 (4%)	*EPCAM**n* = 7 (1.6%)
**Number of CRC**						
**One**	312 (73.4%)	181 (75.7%)	77 (68.8%)	38 (76.0%)	14 (82.4%)	2 (28.6%)
**2 or more**	113 (26.6%)	58 (24.3%)	35 (31.2%)	12 (24.0%)	3 (17.6%)	5 (71.4%)
**Mean age at first CRC ^2^ diagnosis (range)**	47.6 y ^1^ (18–86)	45.2 y ^1^ (18–86)	46.7 y ^1^ (21–83)	56.3 y ^1^ (33–78)	58.8 y ^1^ (38–72)	44.8 y ^1^ (33–61)
**Type of second CRC ^2^**						
**Synchronous**	42 (9.9%)	21 (8.8%)	9 (8.0%)	7 (14%)	2 (11.8%)	3 (42.9%)
**Metachronous**	71 (16.7%)	37 (15.5%)	26 (23.2%)	5 (10%)	1 (5.9%)	2 (28.6%)

632 colorectal cancers were diagnosed in 425 subjects. ^1^
*y* years, ^2^
*CRC* colorectal cancer.

**Table 3 cancers-12-03419-t003:** Number of second colorectal cancer according to mutated gene and type of surgery performed.

Characteristics	TOTAL *n* = 290 (29 ^1^/261 ^2^)	*MLH1**n* = 156 (19 ^1^/137 ^2^)	*MSH2**n* = 83 (5 ^1^/78 ^2^)	*MSH6**n* = 31 (1 ^1^/30 ^2^)	*PMS2**n* = 14 (1 ^1^/13 ^2^)	*EPCAM**n* = 6 (3 ^1^/3 ^2^)
**ONE CRC *n* = 192**						
**Extended surgery**	16/29 (55.2%)	13/19 (68.4%)	3/5 (60%)	0/1 (0%)	0/1 (0%)	0/3 (0%)
**Segmental surgery**	176/261 (67.4%)	92/137 (67.2%)	50/78 (64.1%)	21/30 (70%)	12/13 (92.4%)	1/3 (33.3%)
**SYNCHRONOUS CANCER *n* = 35**						
**Extended surgery**	12/29 (41.4%)	6/19 (31.6%)	2/5 (40.0%)	1/1 (100%)	1/1 (100%)	2/3 (66.7%)
**Segmental surgery**	23/261 (8.8%)	11/137 (8.0%)	7/78 (9.0%)	4/30 (13.3%)	0/13 (0%)	1/3 (33.3%)
**METACHRONOUS CANCER *n* = 63**						
**Extended surgery**	1/29 (3.4%)	0/19 (0%)	0/5 (0%)	0/1 (0%)	0/1 (0%)	1/3 (33.3%)
**Segmental surgery**	62/261 (23.8%)	34/138 (24.8%)	21/78 (26.9%)	5/30 (16.7%)	1/13 (7.7%)	1/3 (33.3%)

Information about surgery was available in 290 subjects: 192 with one CRC, 35 with synchronous CRC and 63 individuals with metachronous CRC. Percentages of type of second CRC are calculated according to the total number of individuals in each surgery group. ^1^ Extended surgery, ^2^ Segmental surgery.

**Table 4 cancers-12-03419-t004:** Number and location of gynecological cancer according to mutated genes.

Characteristics	TOTAL 150/531 (28.2%)	*MLH1* 52/259 (20.1%)	*MSH2* 58/146 (39.7%)	*MSH6* 32/85 (37.6%)	*PMS2* 6/25 (24.0%)	*EPCAM* 2/16 (1.3%)
**Localization of gynecological cancer**						
**Endometrial**	114 (76.0%)	40 (76.9%)	45 (77.6%)	24 (75.0%)	4 (66.7%)	1 (50.0%)
**Ovarian**	27 (18.0%)	9 (17.3%)	10 (17.2%)	5 (15.6%)	2 (33.3%)	1 (50.0%)
**Endometrial + ovarian**	9 (6.0%)	3 (5.8%)	3 (5.2%)	3 (9.4%)	0 (0.0%)	0 (0.0%)
**Mean age at gynecological cancer diagnosis (range)**	49.9 y ^1^ (28–80)	47.6 y ^1^ (31–78)	46.4 y ^1^ (28–80)	51.0 y ^1^ (38–79)	53.5 y ^1^ (42–66)	38.0 y ^1^ (38–38)

150 females with LS developed a gynecological cancer. Percentages of gynecological cancers in each mutated gene group are calculated according to the total number of females in the cohort carrier of each mutated gene. ^1^
*y* years.

**Table 5 cancers-12-03419-t005:** Number of gynecological cancer according to mutated genes and type of surgery performed in all LS women.

Characteristics	TOTAL *n* = 531 (66 ^1^/465 ^2^)	*MLH1**n* = 259 (33 ^1^/226 ^2^)	*MSH2**n* = 146 (19 ^1^/127 ^2^)	*MSH6**n* = 85 (10 ^1^/75 ^2^)	*PMS2**n* = 25 (3 ^1^/22 ^2^)	*EPCAM**n* = 16 (1 ^1^/15 ^2^)
ENDOMETRIAL CANCER *n* = 123						
RRGS ^1^	6 (9.1%)	4 (12.1%)	2 (10.5%)	0 (0%)	0 (0%)	0 (0%)
non-RRGS ^2^	117 (25.2%)	39 (17.3%)	46 (36.2%)	27 (36.0%)	4 (18.2%)	1 (6.7%)
OVARIAN CANCER *n* = 36						
RRGS ^1^	0 (0.0%)	0 (0.0%)	0 (0.0%)	0 (0.0%)	0 (0.0%)	0 (0.0%)
non-RRGS ^2^	36 (7.7%)	12 (5.3%)	13 (10.2%)	8 (10.7%)	2 (9.1%)	1 (6.7%)

^1^*RRGS* risk reducing gynecological surgery, ^2^
*non-RRGS*: non-risk reducing gynecological surgery.

**Table 6 cancers-12-03419-t006:** Studies comparing colorectal cancer incidence and survival between extended surgery and segmental surgery in Lynch syndrome population.

Author	Year	Collected Data/Type of Study	*n* (ES ^1^/SS ^2^)	Population	Follow-Up (Years)	Rate of mCRC ^3^ (ES ^1^/SS ^2^)	10 Years Overall Survival (ES ^1^/SS ^2^)
Vasen [13]	1993	Retrospective Multicentric International	54 (17 ^1^/37 ^2^)	Ams ^4^	5.8 (1–10)	11.8% ^1^ vs. 21.6% ^2^ (*p* = 0.394)	n.r. ^6^
De Vos tot Nederveen WH [14]	2002	Retrospective Multicentric National	97 (29 ^1^/68 ^2^)	LS ^5^ (*MLH1*, *MSH2*, *MSH6*)	ES ^1^: 5 (1–15) SS ^2^: 6.8 (0–15)	3.5 ^1^ vs. 11.8% ^2^ (*p* > 0.05)	n.r. ^6^
Kalady MF [18]	2010	Retrospective Single-institution	296 (43 ^1^/253 ^2^)	Ams ^4^ (11 LS ^5^ confirmed)	8,7 (n.r)	8.0 ^1^ vs. 25.0% ^2^ (*p* = 0.016)	n.r. ^6^
Natarajan N [19]	2010	Retrospective Single-institution	106 (37 ^1,7^/69 ^2^)	LS ^5^ (*MLH1*, *MSH2*)	12 (5–20)	10.8 ^1^ vs. 33.3%^2^ (*p* = 0.006)	86.5 ^1^ vs. 76.8% ^2^ (*p* = 0.239)
Parry S [20]	2011	Retrospective Multicentric International	382 (50 ^1^/332 ^2^)	LS ^5^ (*MLH1*, *MSH2*, *MSH6*, *PMS2*)	ES ^1^: 8 (1–30) SS ^2^: 9 (1–40)	0 vs. 22.3% ^2^ (*p* = 0.019)	98 vs. 97% (*p* = 0.692)
Stupart DA [21] (Stupart et al., 2011)	2011	Retrospective Single-institution	60 (21 ^1^/39 ^2^)	LS ^5^ (*MLH1*, *MSH2*)	ES ^1^: 8 (0–34) SS ^2^: 6 (1–30)	9.5 ^1^ vs. 20.5% ^2^ (*p* = 0.346)	76 ^1^ vs. 62% ^2^ (*p* = 0.222)
Aronson M [22]	2015	Retrospective Single-institution	105 (29 ^1^/76 ^2^)	LS ^5^ (*MLH1*, *MSH2*, *MSH6*, *PMS2*)	6.2 (0–55)	10.3 ^1^ vs 28.9% ^2^ (*p* = 0.071)	n.r. ^6^
Kim TJ [23]	2017	Retrospective Single-institution	106 (30 ^1^/76 ^2^)	LS ^5^ (*MLH1*, *MSH2*, *MSH6*, *EPCAM*)	ES ^1^: 5.7 (1–13) SS ^2^: 6.4 (0–14)	0 ^1^ vs. 17.1% ^2^ (*p* = 0.038)	82.9 ^1^ vs. 83.3% ^2^ (*p* = 0.659) ^9^
Hiatt MJ [24]	2017	Retrospective Single-institution	64 ^8^ (16 ^1^/48 ^2^)	LS ^5^ (*MLH1*, *MSH2*, *MSH6*, *EPCAM*)	n.r. ^6^	6.3 ^1^ vs. 27.0% ^2^ (*p* n.r. ^6^)	81.0 ^1^ vs. 82.8% ^2^ (*p* = 0.471)
Renkonen- Sinisalo L [25]	2017	Retrospective Multicentric National	242 (98 ^1^/144 ^2^)	LS ^5^ (*MLH1*, *MSH2*, *MSH6*)	15.0 (0–32)	5.1 ^1^ vs. 25.0% ^2^ (*p* < 0.001)	47.2 ^1^ vs. 41.1% ^2^ (*p* = 0.83) ^10^
Roh SJ [15]	2020	Retrospective Single-institution	87 (51 ^1^/36 ^2^)	Ams ^4^	ES ^1^: 7.7 (n.r) SS ^2^: 6.6 (n.r)	5.9 ^1^ vs. 2.8% ^2^ (*p* = 0.637)	n.r. ^6^
Heneghan HM [16]	2015	Meta-analysis	948 (168 ^1^/780 ^2^)	LS ^5^ + Ams ^4^	8.9 (5–12)	6.8 ^1^ vs. 23.5% ^2^ (*p* < 0.005)	89.8 ^1^ vs. 90.7% ^2^ (*p* = 0.085)
Anele CC [17]	2017	Meta-analysis	871 (166 ^1^/705 ^2^)	LS ^5^	7.6 (6–12)	6 ^1^ vs. 22.8% ^2^ (*p* < 0.0001)	n.r. ^6^
CURRENT REPORT	2020	Retrospective Single-institution	293 (29 ^1^/264 ^2^)	LS ^5^ (*MLH1*, *MSH2*, *MSH6*, *PMS2*, *EPCAM*)	ES ^1^: 10.9 (0–28) SS ^2^: 14.7 (0–47)	3.4 ^1^ vs. 23.8% ^2^ (*p* < 0.0001)	n.r. ^6^

^1^*ES* extended surgery, ^2^
*SS* segmental surgery, ^3^
*mCRC* metachronous colorectal cancer, ^4^
*Ams* families which meet Amsterdam criteria, ^5^
*LS* Lynch syndrome patients, ^6^
*n.r.* not reported, ^7^ extended right hemicolectomy is included in extended surgery, ^8^ only considered when initial tumor is right-sided tumor, ^9^ 15 years overall survival, ^10^ 25 years overall survival.

**Table 7 cancers-12-03419-t007:** Studies comparing gynecological cancer incidence and survival between risk reducing gynecological surgery and surveillance in Lynch syndrome population.

Author	Year	Collected Data/Type of Study	n (RRGS ^1^/Non-RRGS ^2^)	Follow-Up (Years) (RRGS ^1^/Non-RRGS ^2^)	Rate EC ^3^ (RRGS ^1^/Non-RRGS ^2^)	Rate OC ^4^ (RRGS ^1^/Non-RRGS ^2^)	10 Years Overall Survival (RRGS ^1^/Non-RRGS ^2^)
Schmeler KM [34]	2006	Retrospective Multicentric National (USA)	315 (61 ^1^/254 ^2^)	13.3 ^1^ (0.5–38) 7.4 ^2^ (0.1–35)	0 ^1^ vs. 33.0% ^2^ (*p* < 0.001)	0 ^1^ vs. 5.5% ^2^ (*p* = 0.09)	n.r. ^6^
CURRENT REPORT	2020	Retrospective Single-institution	531 (66 ^1^/465 ^2^)	8.7 ^1^ (0–43) 10.4 ^2^ (0–45)	9.1 ^1^ vs. 25.2% ^2^ (*p* < 0.001)	0 ^1^ vs. 7.7% ^2^ (*p* N/A ^5^)	n.r. ^6^

^1^*RRGS* risk reducing gynecological surgery, ^2^
*non-RRGS* non- risk reducing gynecological surgery, ^3^
*EC* Endometrial cancer, ^4^
*OC* Ovarian cancer, ^5^
*N/A* not assessable, ^6^
*n.r.* not reported.

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
