# Peer review of "Assessing Effectiveness of Colonic and Gynecological Risk Reducing Surgery in Lynch Syndrome Individuals"

_cancers, 2020, doi:10.3390/cancers12113419_

Round 1

Reviewer 1 Report

It has been my pleasure to read this paper for its methods, findings, and presentation.

The Introduction is very good, I have no suggestions to make there.

The Results are generally well presented but I think a little extra work on the tables might make them more readable and I would want to be sure the figures are ultimately produced in higher quality than they are currently (perhaps also adding confidence bands and/or number at risk table).

The Methods are well described apart from it not being clear for risk-reducing gynaecological surgery how participants are analysed. I think somebody can only enter the RRGS "arm" if they receive RRGS, i.e., do not have a diagnosis of gynaecological cancer, but what happens to the time between them starting to be observed and them receiving RRGS? If they are counted within the RRGS arm then this will lead to immortal time bias. If not, why are we losing their data and it not contributing to the "non-RRGS" arm? And if they do contribute to both arms, is there any accounting for the dependence in the statistical analysis?

Also I have some concerns that the decision for extended versus segmental colectomy will be affected by confounders, particularly the extent/stage of disease at time of surgery, and the period in which decisions were taken. I would be happy for additional sensitivity/subgroup analyses to be presented in supplementary materials, accepting that these will likely be underpowered.

Author Response

Reviewer # 1 comments to the author and author answers

It has been my pleasure to read this paper for its methods, findings, and presentation.

The Introduction is very good, I have no suggestions to make there.

The Results are generally well presented but I think a little extra work on the tables might make them more readable and I would want to be sure the figures are ultimately produced in higher quality than they are currently (perhaps also adding confidence bands and/or number at risk table).

We agree with the reviewer that tables could be more readable and figures at higher quality. For this reason we have improved the format of all of them.

The Methods are well described apart from it not being clear for risk-reducing gynaecological surgery how participants are analysed. I think somebody can only enter the RRGS "arm" if they receive RRGS, i.e., do not have a diagnosis of gynaecological cancer, but what happens to the time between them starting to be observed and them receiving RRGS? If they are counted within the RRGS arm then this will lead to immortal time bias. If not, why are we losing their data and it not contributing to the "non-RRGS" arm? And if they do contribute to both arms, is there any accounting for the dependence in the statistical analysis?

The individuals included in the RRGS arm are the ones who undergo a risk-reducing or prophylactic gynaecological surgery and who have not been diagnosed with gynaecological cancer prior to that surgery. The rest of the women in the series are included in the non-RRGS group.

Also I have some concerns that the decision for extended versus segmental colectomy will be affected by confounders, particularly the extent/stage of disease at time of surgery, and the period in which decisions were taken. I would be happy for additional sensitivity/subgroup analyses to be presented in supplementary materials, accepting that these will likely be underpowered.

We agree with reviewer 1 that the decision of extended vs. segmental colectomy can be affected by confounders, as patients were allocated to each surgery group based on the surgeon decision. At the time of first CRC surgery most individuals were not known to have Lynch syndrome so the decision was based on clinical characteristics (stage and localization of the tumor, existence of synchronous CRC) and patient characteristics (age at CRC, expected bowel function, personal preferences, etc). We applied a test of comparison of proportions between groups and we can state that there are no statistical significant differences between groups regarding gender, mutated gene, age at first CRC and CRC stage (information available at Table 1, Table 2, Table 3 and Table S1).

This limitation is discussed in the discussion section (lines 357-362).

Reviewer 2 Report

This is an interesting analysis on patients with Lynch syndrome which takes into consideration gynecological and general surgical aspects. The work is well done, and the sample is adequate.

I would ask the authors for some small adjustments.

In introduction line 72 I think that MSH2 promoter methylation is a mistake

In table 4 if it is possible to add the histology of the gynecological tumors.

In discussion:

Do you have any indications for a combined RR surgery between the colon and the genital system? Any data?

Although it is not the role of the article, but recently the prognostic problem of somatic mutations of the MMR genes especially in gynecological tumors is engaging the scientific world. In a recent publication (Dondi et al. Int J Mol Sci. 2020) the authors raised the issue; do you have any suggestions to put in the discussion section?

Author Response

Reviewer # 2 comments to the author and author answers

This is an interesting analysis on patients with Lynch syndrome which takes into consideration gynecological and general surgical aspects. The work is well done, and the sample is adequate.

I would ask the authors for some small adjustments.

In introduction line 72 I think that MSH2 promoter methylation is a mistake

As far as we know, germline deletions at the 3'-end of EPCAM eliminate the transcription termination signal inducing the expression of a EPCAM–MSH2 fusion transcript, resulting in silencing MSH2 by the hypermethylation of its promoter. MSH2 methylation level correlates with EPCAM expression levels, usually high in those epithelial tissues in which EPCAM is expressed.

We have tried to clarify the sentence in lines as 71-73:

“Lynch Syndrome (LS) is characterized by an inherited defect in the mismatch repair (MMR) genes ((MLH1,MSH2,MSH6,PMS2) or EPCAM gene deletions, resulting in silencing MSH2 gene in epithelial tissues).”

References:
- Ligtenberg MJ, Kuiper RP, Chan TL et al. Heritable somatic methylation and inactivation of MSH2 in families with Lynch syndrome due to deletion of the 3’ exons of TACSTD1. Nat Genet 2009: 41: 112–117. 

- Kovacs ME, Papp J, Szentirmay Z, Otto S, Olah E. Deletions removing the last exon of TACSTD1 constitute a distinct class of mutations predisposing to Lynch syndrome. Hum Mutat 2009: 30: 197–203

In table 4 if it is possible to add the histology of the gynecological tumors.

We have added this information in Table S3. We agree with reviewer 2 that this information is relevant and important. However, we feel that adding this information to table 4 would make the table more difficult to understand and less readable

In discussion:

Do you have any indications for a combined RR surgery between the colon and the genital system? Any data?

In the latest publication of the Manchester International Consensus Group they recommend that “The Consensus Group recommends that risk-reducing colorectal and gynecological surgery is carried out at the same time, when indicated and where posible (grade C)”. As fas as we know, any publication has evaluated this topic in a retrospective or a prospective analysis and this recommendation is based on expert opinions to try to avoid or reduce morbility and mortality associated with the surgical act.

This recommendation is included in our manuscript in lines 384-386

Reference:

- Crosbie EJ, Ryan NAJ, Arends MJ, Bosse T, Burn J, Cornes JM, Crawford R, Eccles D, Frayling IM, Ghaem-Maghami S, Hampel H, Kauff ND, et al. The Manchester International Consensus Group recommendations for the management of gynecological cancers in Lynch syndrome. Genet Med 2019;0:1–11.

Although it is not the role of the article, but recently the prognostic problem of somatic mutations of the MMR genes especially in gynecological tumors is engaging the scientific world. In a recent publication (Dondi et al. Int J Mol Sci. 2020) the authors raised the issue; do you have any suggestions to put in the discussion section?

Endometrial cancers in Lynch syndrome have shown to have good prognosis (10y overall survival of 89%) (ref 1). In this publication commented by reviewer 2, 5 years overall survival in Lynch-like endometrial cancers seem to be much worse than endometrial cancer in LS patients (91-92% vs 100%). We speculate that this can be due to different carcinogenic processes and activation of different pathways, even though MMR deficiency is present in both type of tumours. Moreover, Lynch-like syndrome individuals are generally not identified until the existence of the cancer and therefore are not included in surveillance programs or offered risk reducing surgeries.

We agree with reviewer 2 that this topic is of great importance and it should be thoroughly studied. However, we believe that this is not the scope of our study and it could cause confusion if we include the discussion of this topic in our manuscript.

Reference:
1. Dominguez-Valentin M, Sampson JR, Seppälä TT, ten Broeke SW, Plazzer J-P, Nakken S, Engel C, Aretz S, Jenkins MA, Sunde L, Bernstein I, Capella G, et al. Cancer risks by gene, age, and gender in 6350 carriers of pathogenic mismatch repair variants: findings from the Prospective Lynch Syndrome Database. Genet Med 2020;22:15-25

Reviewer 3 Report

The manuscript by Duena et al. reports a single center retrospective investigation on the role of surgery in reducing risk and mortality for colorectal (CRC) and gynaecological (GC) cancer in Lynch syndrome. Few well-conducted studies are available in the literature on this topic, and the present study is particularly welcomed.

The data presented seem to suggest an effect of extended surgery in reducing the risk of CRC, but not mortality. For GC surgery reduces risk and mortality, though in this case surgery is considered against surveillance. These results are somewhat expected, however they are based on a sufficient number of observations and on a wide follow-up time.

Three points are critical:

  1. The retrospective design of the study of course does not allow to exclude bias in the decision of operation against surveillance, though authors say that there are not differences between groups for that allocation.
  2. Furthermore, the suggeston of giving chemopreventive drugs to patients operated on of segmental resection for CRC is questionable, because at present no data are available on this point.
  3. Moreover, the side-effects of surgery on the quality of life are somewhat overlooked, and the authors in the discussion section seem to rely only on other works in the literature. At present the decision of going to surgery is based on a multidisciplinary approach, taking into account the patients’ view. I suggest authors to better discuss these points.

As a whole I think that the paper adds an important piece of knowledge on the effect of surgery in gene-carriers of mutations responsible of Lynch syndrome, expecially those affected by GC.

Author Response

Reviewer # 3 comments to the author and author answers

The manuscript by Duenas et al. reports a single center retrospective investigation on the role of surgery in reducing risk and mortality for colorectal (CRC) and gynaecological (GC) cancer in Lynch syndrome. Few well-conducted studies are available in the literature on this topic, and the present study is particularly welcomed.

The data presented seem to suggest an effect of extended surgery in reducing the risk of CRC, but not mortality. For GC surgery reduces risk and mortality, though in this case surgery is considered against surveillance. These results are somewhat expected, however they are based on a sufficient number of observations and on a wide follow-up time.

Three points are critical:

  1. The retrospective design of the study of course does not allow to exclude bias in the decision of operation against surveillance, though authors say that there are not differences between groups for that allocation.

We agree with reviewer 3 that the retrospective design of the study can induce to bias when selecting patients for RRGS vs. surveillance. After 2015, RRGS was offered to all the women who fulfilled criteria (older than 35-40 years of age and who had completed childbearing) and before 2015 this intervention was considered individually and offered to women who fulfilled the same criteria and who had an important family history of gynaecological cancer or had to be intervened for any gynaecological benign lesion. Each patient made the final decision of the intervention, so there are multiple factors in this decision that can lead to bias. We applied a test of comparison of proportions between groups and we can state that there were no statistical significant differences between groups regarding age at gynaecological surgery or proportion of carriers for each gene (p<0.5). However, differences existed regarding number of patients included and time of follow-up.

This limitation is discussed in the results section (lines -----) and the discussion section (lines 367-369).

  1. Furthermore, the suggeston of giving chemopreventive drugs to patients operated on of segmental resection for CRC is questionable, because at present no data are available on this point.

We agree with reviewer 3 that chemoprevention after colonic surgery has not been evaluated. For that purpose and to avoid confusions, we have changed this sentence in the discussion section to “To date, no study has compared extended surgery vs. segmental surgery plus chemoprevention with aspirin, so currently we cannot conclude that one approach is better than the other” (lines 258-260).

  1. Moreover, the side-effects of surgery on the quality of life are somewhat overlooked, and the authors in the discussion section seem to rely only on other works in the literature. At present the decision of going to surgery is based on a multidisciplinary approach, taking into account the patients’ view. I suggest authors to better discuss these points.

We agree with reviewer 3 that to recall information in our series about QoL would have been of great interest. However, this information was not collected. The existing literature has not found relevant differences in generic quality of life (You N et al, 2008; Haanstra JF et al, 2012). We speculate that the same may apply to our population.

Even though this matter was partially discussed in the discussion section, we have expanded the discussion with the reviewer’s comments (lines 386 - 388).

As a whole I think that the paper adds an important piece of knowledge on the effect of surgery in gene-carriers of mutations responsible of Lynch syndrome, expecially those affected by GC.

We kindly thank reviewer 3 for his/her comments.
